# Efficiently Verifiable Proofs of Data Attribution

Ari Karchmer[*]        Martin Pawelczyk[†]        Seth Neel[‡]

## Abstract

Data attribution methods aim to answer useful counterfactual questions like "what would a ML model's prediction be if it were trained on a different dataset?" However, estimation of data attribution models through techniques like empirical influence or "datamodeling" remains very computationally expensive. This causes a critical trust issue: if only a few computationally rich parties can obtain data attributions, how can resource-constrained parties trust that the provided attributions are indeed "good," especially when they are used for important downstream applications (e.g., data pricing)? In this paper, we address this trust issue by proposing an interactive verification paradigm for data attribution. An untrusted and computationally powerful *Prover* learns data attributions, and then engages in an *interactive proof* with a resource-constrained *Verifier*. Our main result is a protocol that provides formal *completeness*, *soundness*, and *efficiency* guarantees in the sense of Probably-Approximately-Correct (PAC) verification (Goldwasser et al., 2021). Specifically, if both Prover and Verifier follow the protocol, the Verifier accepts data attributions that are $\varepsilon$-close to the optimal data attributions (in terms of the Mean Squared Error) with probability $1 - \delta$. Conversely, if the Prover arbitrarily deviates from the protocol, even with *infinite* compute, then this is *detected* (or it still yields data attributions to the Verifier) except with probability $\delta$. Importantly, our protocol ensures the Verifier's workload, measured by the number of independent model retrainings it must perform, scales only as $O(1/\varepsilon^2)$; i.e., *independently* of the dataset size. At a technical level, our results apply to efficiently verifying any linear function over the boolean hypercube computed by the Prover, making them broadly applicable to various attribution tasks.

## 1 Introduction

The attempt to understand and explain the behavior of complex machine learning systems has given rise to the field of *attribution*. Broadly, attribution methods seek to trace model outputs or behaviors back to their origins. This takes several forms. One prominent form is *training data attribution*, which aims to quantify the influence of individual training examples on model predictions, facilitating interpretability, debugging, and data valuation (e.g., Koh and Liang, 2017; Ghorbani and Zou, 2019; Ilyas et al., 2022). Complementary to this, *component attribution* (Shah et al., 2024) focuses on decomposing a model's prediction in terms of its internal components, such as convolutional filters or attention heads, to understand how they combine to shape model behavior.

While many applications of attribution may be of interest to the model developers, others are inherently of interest to third parties who may not have access to the internal details of how the attributions were computed. For instance, consider a framework where entities that supply data for training are compensated in proportion to the degree their training data influences model outputs, an idea which

---

[*]Morgan Stanley Machine Learning Research, Harvard Business School, `akarchmer0@gmail.com`

[†]Harvard Business School, `martin.pawelczyk.1@gmail.com`

[‡]Google Research, Harvard Business School, `sethneel@google.com`

has been extensively discussed (Ghorbani and Zou, 2019; Castro Fernandez, 2023; Jia et al., 2023; Choe et al., 2024). In this setting, a user $i$ may not only be concerned that the attributions for the dataset as a whole are accurate—meaning they have low predictive error in absolute terms—but that if the ground truth attribution for user $j$ is lower than for user $i$, the received attributions also satisfy this property. As another example, suppose a doctor is using a model to provide diagnoses, and wants to understand what training data the model used to reach its conclusions, in order to sanity check them.

In each of these settings, rather than simply trusting that attributions are computed correctly, third parties may want to *verify* that the attributions are "right," before making important downstream decisions. The challenge arises because of a computational disparity—third parties using these attributions typically have *far, far fewer* computational resources than the model developer that is providing them, whether they are an individual or even another small lab in academia or industry. Thus, given the large computational expense associated with estimating ground truth attributions, we have the following motivating question:

> *Is it possible to design a protocol by which a computationally-limited third party*
> *can "verify" the correctness of attributions?*

**Our Contribution, in a Nutshell.** In this paper, we demonstrate an interactive two-message protocol where the third party's computational cost, which is measured by the number of full model retrainings it must conduct, is independent of the attribution set size $N$ (e.g., in data attribution $N$ is dataset size, while in component attribution it is the number of components). That is, this number of required retrainings does not grow as a function of $N$. Our protocol comes with strong guarantees of correctness for the third party, even if the model developer deviates from the agreed protocol arbitrarily, and with *infinite* compute. To make this precise, we must specify (i) the form of attributions we study, (ii) how we measure computational cost, (iii) what our accuracy metric is, and finally (iv) what guarantees of correctness we can (*and should*) provide for our protocol.

While we will formalize each of these in section 2, we discuss each of them here in order to clearly state our contributions. Additionally, while our work applies to data attribution methods in general, we will proceed with predictive data attribution in mind, for purposes of exposition.

**Predictive Training Data Attribution.** In predictive training data attribution (TDA), the goal is to answer counterfactual questions: "What would have happened to a model's behavior if we had trained it on a different subset of the data?" To formalize this, we consider a *model output function*, $f : \{\pm 1\}^N \to \mathbb{R}$. The domain of $f$ are subsets $x$ of a fixed training dataset $S$ of size $N$ (where $x \in \{\pm 1\}^N$ and $x_i = 1$ indicates inclusion of the $i^{th}$ data point from $S$ and $x_i = -1$ indicates exclusion). The model output function $f(x)$ measures the (expected) model behavior if it were trained on $S$, and is evaluated by retraining a model $\theta \sim \mathcal{T}(x)$ via a training algorithm $\mathcal{T}$, and then computing some function of $\theta$, often the prediction $\theta(z)$ on an input point $z$ (Ilyas et al., 2022), or the test error of $\theta$ on some subgroup of the data (Jain et al., 2024).

The objective of TDA is then to construct a *predictive model*, often a linear function $g(x) = \langle \mathbf{a}, x \rangle$, where $\mathbf{a}$ are the *attribution scores*, in order to predict the *counterfactual effect* of training on a different subset of the data. The quality of this predictive model can be measured by its Mean Squared Error (MSE) with respect to the true model output function $f$ and a binomial product distribution $\mathcal{B}_p$ over the subsets:

$$\text{mse}^{(p)}(f, \langle \mathbf{a}, \cdot \rangle) \triangleq \mathop{\mathbb{E}}_{x \sim \mathcal{B}_p} \left[ (f(x) - \langle \mathbf{a}, x \rangle)^2 \right].$$

Achieving a low MSE indicates that the attribution scores $\mathbf{a}$ accurately predict how different subsets of data affect the model's behavior.[4]

**Remark 1.** *While we will focus on the well-studied area of TDA throughout, the recently proposed "component attribution," (Shah et al., 2024) that aims to determine which internal components of the model are primarily responsible for specific predictions, also fits neatly into our framework. In this case, $S$ defined above corresponds to the set of total model components, sampling $x \sim \mathcal{B}_p$ corresponds to sampling a subset of components, and the model output function $f$ consists of computing a forward*

---

[4]Note, $f$ is technically a randomized function, but we will assume a fixed random seed, and that $f$ becomes deterministic once the random seed is fixed.

*pass on a test input z where activations from components $\notin x$ are set to a constant. The methods in this paper can also be used to verify the accuracy of model component attributions.*

**Specific Methods for Training Data Attribution.** There are two major classes of algorithms for TDA: those based on retraining, and gradient-based methods like influence functions or representer point methods that leverage approximations to the model training process or architecture. In retraining based methods, first samples $x_i \subset S$ are drawn (typically from a product distribution), and the output function $f(x_i)$ is computed, which involves training a model $\theta(x_i)$ and then computing the relevant statistic from $\theta(x_i)$. Different retraining-based attribution methods generally correspond to different techniques for estimating $\mathbf{a}$ from samples $\{(x_i, f(x_i))\}_{i=1}^M$ computed by retraining. In Datamodels (Ilyas et al., 2022), $\mathbf{a}$ is computed via a Lasso regression to encourage sparsity in $\mathbf{a}$, where the model output function computes the difference in correct class logit for a specific test input $z$. In Empirical Influence (Feldman and Zhang, 2020), the $j^{th}$ coefficient $\mathbf{a}_j$ is estimated as the average of the sampled outcomes where the $j^{th}$ datapoint was included: $M^{-1} \sum_{x_i \in M_j} f(x_i)$, where $M_j = \{x_i : (x_i)_j = 1\}$. The clever work of Saunshi et al. (2022) proves that these two methods are essentially theoretically equivalent, though performance can vary in practice, in part due to optimizations such as applying the LASSO algorithm.

Most gradient-based TDA methods are based on approximations to the continuous influence function (Law, 1986; Koh and Liang, 2017; Park et al., 2023; Grosse et al., 2023; Guu et al., 2023). While these methods are more computationally efficient than exact retraining, the assumptions underlying influence functions (e.g., convexity, training to convergence) often do not hold in deep learning, potentially undermining their suitability for data attribution (Bae et al., 2022). Yet another class of TDA methods aim to approximate the model with a simpler model class, where the attributions can be computed analytically (Park et al., 2023; Yeh et al., 2018). Prior work has found that retraining methods can be more accurate than those based on influence functions or model approximations, in settings where they are both tractable to compute (Ilyas et al., 2022), but so far these retraining based methods are unable to scale to large models (Grosse et al., 2023).

**Computational Cost.** While highly accurate retraining-based predictive data attribution can prove invaluable for tasks like dataset selection (Engstrom et al., 2024) and sensitivity analysis (Broderick et al., 2020; Pawelczyk et al., 2023), they come with a large computational cost. Computing a fresh sample $f(x_i)$ corresponds to *retraining an entirely new model from scratch*, and the number of samples required to obtain accurate Datamodels (attributions) is large: the work of Ilyas et al. (2022) trains *three million* ResNet-9 networks on random subsets of the CIFAR-10 dataset to construct their attribution. The recent work by Georgiev et al. (2024) uses Datamodels for the less stringent task of machine unlearning, and still requires training $> 10,000$ models. Methods based on the influence function avoid retraining but still require computations like inverse-Hessian-vector products, which can be intractable for very large models. Moreover, even methods based on influence functions may require ensembling over multiple training runs to achieve reasonable accuracy (Park et al., 2023).

**Trust Issues: Naive Verification Fails.** The computational barriers described above put TDA, especially retraining-based methods like Datamodels or Empirical Influence, beyond the reach of individual practitioners or even moderately resourced academic labs. This effectively centralizes the task within a few industrial or institutional powerhouses.

Centralization causes trust issues. If only a few parties can effectively perform attribution, then, given a candidate attribution $\mathbf{a}$ computed by the computationally powerful group, how can a third party efficiently establish the accuracy of $\mathbf{a}$? In keeping with the rich body of work relating to interactive proofs, going forward we will refer to the computationally powerful party as the *Prover*, and the resource-constrained third party as the *Verifier*.

A naive idea that seemingly fits into our setting, is for the Verifier to try to check the MSE of any proposed attributions $\mathbf{a}$.

- The Verifier randomly samples a collection of subsets $T \subset \{\pm 1\}^N$ of size $m$, obtains $f(x) \ \forall x \in T$, and uses that data to estimate $\mathrm{mse}^{(p)}(f, \langle \mathbf{a}, \cdot \rangle) \approx \frac{1}{m} \sum_{x_i \in T} (\langle \mathbf{a}, x_i \rangle - f(x_i))^2$.

When $m$ is proportional to $1/\varepsilon^2$, the estimate is within an additive $\varepsilon$ error from the true MSE with high probability. Hence, the Verifier *can* check that the Prover has given him attributions which have

MSE at most $\alpha + \varepsilon$, for some pre-determined $\alpha$. Most importantly, the Verifier can do this using only $O(1/\varepsilon^2)$ retraining procedures on the model—which is *independent* of the data set size $N$!

However, this is *not* actually satisfactory. To see why, consider again the setting of data valuation. For concreteness, imagine each data point is contributed by a user, under the agreement they will be compensated in proportion to the attribution scores for the data they contribute. Now, suppose the Prover proposes candidate attributions $\mathbf{a}$ such that $\mathrm{mse}^{(p)}(f, \langle \mathbf{a}, \cdot \rangle) = 0.22$. Then, when the Verifier checks the MSE, they will obtain an estimate $\alpha' \in [0.22 - \varepsilon, 0.22 + \varepsilon]$ with high probability. But, what if there exists $\mathbf{a}'$ such that $\mathrm{mse}^{(p)}(f, \langle \mathbf{a}', \cdot \rangle) = 0.022$?

Indeed, in order to reduce total costs, a *malicious* Prover might cheat as follows:

- Compute $\mathbf{a}$ by brute-force Empirical Influence.[5]
- Use $\mathbf{a}' = \mathbf{a} \cdot \frac{1}{2}$ (scalar multiplication).

Recall that in our simple data valuation setting, the Prover will pay out data sellers proportional to their attribution. Thus, by dividing the true attributions by 2, the Prover will save 50% money (at the necessary cost of increase to MSE).

As in the naive protocol, only checking the MSE fails to shed light on whether such cheating has been carried out by the Prover (without loss of generality on the exact cheating strategy). To emphasize the point further, consider the case that the Prover wants to advantage a subset of data contributors $A$ over all other contributors, so they receive more payment. In order to do so, they modify $\mathbf{a}$ by increasing coordinates where $i \in A$ by some value $\beta > 0$ to produce a new attribution vector $a'$. If $\beta$ is reasonably small, the modified $a'$ might also have low MSE, and so a Verifier that merely checks the MSE is below a certain threshold would accept these attributions.

It is not a priori clear whether or not the Verifier has any way of defending itself against this form of cheating Prover, without having to compute the attributions itself. That is, without having to do the work of the Prover.

**A Better Notion of Verifiability.** As we have hopefully demonstrated, we need a more meaningful notion of verifiability, that goes beyond simply checking the MSE of the proposed attribution. To this end, we suggest estimating *sub-optimality*. Put simply, the Verifier should ensure the attribution is "$\varepsilon$-close" to optimal. This can be written as verifying the fact that

$$\mathrm{err}(\mathbf{a}', \Phi(S)) = \mathrm{mse}^{(p)}(f, \langle \mathbf{a}', \cdot \rangle) - \mathrm{mse}^{(p)}(f, \langle \Phi(S), \cdot \rangle) \leq \varepsilon. \tag{1}$$

Here, $\Phi(S)$ denotes the *optimal* attribution vector for the set $S$, with respect to MSE. That is,

$$\Phi(S) \triangleq \arg\min_{\mathbf{a}} \mathrm{mse}^{(p)}(f, \langle \mathbf{a}, \cdot \rangle). \tag{2}$$

And so, in other words, verifying sub-optimality means checking if the error gap (equation 1) between the proposed attributions, and the optimal attributions, is small.

All in all, checking low sub-optimality tells the Verifier that for this specific setting, the provided attributions are nearly optimal, and therefore the Prover has acted in good faith. Thus, we propose to aim for the following (still informal) guarantees.

**Informal verification Guarantees.** When designing a verification protocol we will obtain:

- **Completeness.** If both Prover and Verifier follow the protocol, the Verifier obtains attribution scores that are **approximately optimal** with respect to predictive MSE, with high probability.

- **Soundness.** If the Prover deviates from the protocol in any way, then, with high probability, the Verifier either outputs "abort" or still obtains attribution scores that are **approximately optimal** with respect to predictive MSE.

- **Efficiency.** The Verifier's workload scales independently of the dataset size $N$.

**Remark 2.** *In practice, even an honest, powerful Prover might only compute an estimate $\hat{\mathbf{a}}^\star$ of $\Phi(S)$ due to computational constraints (e.g., using an influence function). The soundness guarantee*

---

[5]It is strongest to consider a malicious Prover to be computationally *unbounded*—as is customary in the theory of interactive proofs Goldwasser et al. (2019, 2021) or statistical zero knowledge (Vadhan, 1999), and others.

*still holds relative to the true $\Phi(S)$. However, the completeness guarantee might be affected if the honest Prover's estimate $\hat{a}^\star$ is itself far from $\Phi(S)$. If $\text{err}(\hat{a}^\star, \Phi(S))$ is inherently large due to the Prover's own estimation limitations, the Verifier might reject even an honest Prover if $\varepsilon$ is set too small. Therefore, the choice of $\varepsilon$ in practice should reflect not only the Verifier's desired precision but also potentially incorporate a tolerance for the best achievable estimation error by an efficient (though powerful) honest Prover.*

## 1.1 Our Contributions

**Conceptual contributions.** Conceptually, this work introduces the idea, motivation, and goal of efficient interactive verification of data attribution.

As we will see in the next section, our formalization of this goal is done via direct connection to the interactive PAC-verification framework (Definition 1) of Goldwasser et al. (2021). As we have discussed, a key challenge in verifying a proposed attribution vector $a'$ is comparing its predictive quality against the *optimal* linear predictor $\Phi(S)$ without actually computing $\Phi(S)$. The PAC-Verification paradigm turns out to be well-equipped to handle this challenge.

**Technical contributions.** As for technical contributions, we demonstrate two efficient protocols for the task of efficiently verifying optimality of proposed attributions.

For the first protocol, it turns out that recent work by Saunshi et al. (2022) actually provides a "residual estimation" algorithm for exactly this task. More specifically, their algorithm estimates the optimal residual (that is, $\text{mse}^{(p)}(f, \langle \Phi(S), \cdot \rangle)$) using only $O(1/\varepsilon^3)$ samples of the function $f$ (Theorem 2), *without* needing to compute $\Phi(S)$ itself. This naturally suggests a potential *non-interactive* verification protocol where:

- The Prover would compute Empirical Influence attributions $a'$ and send them to the Verifier.

- The Verifier could independently estimate the optimal $\text{mse}^{(p)}(f, \langle \Phi(S), \cdot \rangle)$ (within an additive factor of $\varepsilon$) using the algorithm of Saunshi et al. (2022), and also *independently* estimate $\text{mse}^{(p)}(f, \langle a', \cdot \rangle)$ (within an additive factor of $\varepsilon$) using $O(1/\varepsilon^2)$ samples. The Verifier would then accept if the two estimates were $\varepsilon$-close.

The complexity of this non-interactive approach would be dominated by the residual estimation step, resulting in a Verifier cost of $O(1/\varepsilon^3)$ (derived from the sample complexity guarantee of Saunshi et al. (2022)). We consider this non-interactive protocol in detail in section 3.

**Main Technical Contribution.** Our main technical contribution is to build out the above non-interactive protocol into an *interactive* protocol. Our interaction strategy serves to reduce the Verifier's overall complexity to $O(1/\varepsilon^2)$, and is implemented by a "spot-checking" mechanism. The function of the "spot-checking" mechanism is to further move some of the Verifier's work required for residual estimation, onto the prover. This mechanism necessarily requires an interaction, though we only use 2 messages (first the Verifier speaks, and then the Prover responds). To prove the intended qualities of the spot-checking mechanism, we do a non-black box analysis of the proof of the residual estimation algorithm of Saunshi et al. (2022), in order to demonstrate a degree of adversarial robustness in the procedure.

We outline our protocol in Section 4. The interactive protocol satisfies the completeness and soundness guarantees that we have discussed. With respect to efficiency, we improve upon the non-interactive protocol by obtaining a Verifier workload that scales as $O(1/\varepsilon^2)$.

## 2 Formal PAC-verification Framework for Attribution and Main Theorem

To this point, we have not formalized what we mean by a Prover-Verifier protocol for data attribution. Let us begin with formalizing the interaction model.

**Communication.** As part of the interaction, we assume an asynchronous, reliable channel between the parties. An *interaction* consists of a finite sequence of messages $w_1, w_2, \ldots, w_t \in \{0, 1\}^*$, sent alternately between the **Verifier** V and the **Prover** P. Messages are unrestricted bit-strings and may encode, for example, hashes, sketches, model weights, or random seeds.

**Shared resources.** Both the Prover and the Verifier will agree on certain information regarding the objective of the protocol. For instance, the protocol is executed with respect to a fixed training set $S$ (with $|S| = N$), a fixed model architecture, and a fixed objective function for model training which defines the model output function $f : \{\pm 1\}^N \to \mathbb{R}$, as introduced earlier.[6]

**Termination.** After the last message V halts and outputs a single value $\hat{\mathbf{a}} \in \{\text{abort}\} \cup \mathbb{R}^N$, interpreted respectively as rejection or an accepted vector of attribution scores. No further interaction occurs once abort or $\hat{\mathbf{a}}$ is produced.

## 2.1 The Protocol Guarantees

We are now ready to formalize our desired protocol guarantees. Namely, Completeness, Soundness, and Efficiency. To this end, we adapt a previous formalization of a similar setting called Probably-Approximately-Correct Verification (PAC-Verification) (Goldwasser et al., 2021). After introducing the formalism, we will also comment on why it is necessary to use this framework, instead of other formalisms from Cryptography, such as Delegation of Computation protocols (see e.g., Goldwasser et al. (2015)).

Let $\mathcal{X}$ be the space of training examples. Recall the model output function $f : \{\pm 1\}^N \to \mathbb{R}$, which maps a representation of a training subset to a model behavior, and our goal to find an attribution vector $\mathbf{a}$ such that the linear predictor $\langle \mathbf{a}, \cdot \rangle$ has low $\text{mse}^{(p)}(f, \langle \mathbf{a}, \cdot \rangle)$.

Let $\Phi(S) \in \mathbb{R}^N$ denote the optimal attribution vector with respect to MSE.

Our error function $\text{err}(\mathbf{a}, \Phi(S))$ measures the sub-optimality of a candidate attribution vector $\mathbf{a}$ in terms of its predictive MSE performance compared to $\Phi(S)$:

$$\text{err}(\mathbf{a}, \Phi(S)) \triangleq \text{mse}^{(p)}(f, \langle \mathbf{a}, \cdot \rangle) - \text{mse}^{(p)}(f, \langle \Phi(S), \cdot \rangle).$$

Note that $\text{err}(\mathbf{a}, \Phi(S)) \geq 0$. The Verifier's goal is to accept $\mathbf{a}$ if this error gap is less than or equal to an accuracy threshold $\varepsilon$.

Finally, fix a Verifier *cost function* $\kappa : (0, 1)^2 \to \mathbb{N}$.

**Definition 1** (PAC-verification for data attribution, adapted from Goldwasser et al. (2021)). *Fix accuracy $\varepsilon \in (0, 1)$ and confidence $\delta \in (0, 1)$. An **interactive proof system** (V, P) is an $(\varepsilon, \delta)$-PAC verifier for $\Phi$ if, for every dataset $S$, the following hold after V and P exchange messages, and then V outputs a value $\hat{\mathbf{a}} \in \{\text{abort}\} \cup \mathbb{R}^N$:*

*Completeness. If P abides by the protocol, then*

$$\Pr\left[\hat{\mathbf{a}} \neq \text{abort} \ \wedge \ \text{err}\left(\hat{\mathbf{a}}, \Phi(S)\right) \leq \varepsilon\right] \ \geq \ 1 - \delta.$$

*Soundness. For every (possibly computationally unbounded) dishonest prover P′,*

$$\Pr\left[\hat{\mathbf{a}} \neq \text{abort} \ \wedge \ \text{err}\left(\hat{\mathbf{a}}, \Phi(S)\right) > \varepsilon\right] \ \leq \ \delta.$$

*Efficiency. There exists a constant $k > 0$, such that $\kappa(\varepsilon, \delta) < (1/\varepsilon \cdot \log(1/\delta))^k$.*

The Verifier's cost is required to be *independent* of $N = |S|$. For our choices of $\Phi$ and err (as defined above), the honest Prover's cost can be shown to necessarily grow with $N$. Hence, our protocols allow the Verifier to expend arbitrarily less cost than the Prover.

**Why PAC-Verification and not Delegation of Computation?** One might initially consider employing general-purpose cryptographic protocols for delegation of computation or verifiable computation to ensure the prover P performed the expensive attribution computation correctly. However, such approaches are insufficient for the verification goal central to our work and discussed by Goldwasser et al. (2021). First of all, computational overhead for the Prover can be significant in existing cryptographic solutions; this already renders the approach very impractical in our setting, since the honest Prover may already be pushing the limits of their computational power. Second, Cryptographic

---

[6]Later in the paper, we will assume that $f : \{\pm 1\}^N \to [-b, b]$ has a bounded range. This assumption is supported empirically by typical model output functions considered in the literature. For instance: change in correct-class margin at a specific test point (Ilyas et al., 2022).

delegation can typically only ensure that P *executed a specific computation as promised*, given some inputs. It cannot, in general, provide guarantees about the *statistical quality* of the output, particularly whether the resulting attribution scores $\mathbf{a}'$ are indeed $\varepsilon$-close to the optimal scores $\Phi(S)$ according to our error metric $\mathrm{err}(\cdot, \cdot)$. We refer to section F for a continued discussion.

## 2.2 Main Theorem

The main result of this paper is an $(\varepsilon, \delta)$-PAC verifier for data attribution, where correctness is measured by the MSE difference defined in $\mathrm{err}(\cdot, \cdot)$. At this point, we only need to define the Verifier cost function in order to state the theorem.

We will take the Verifier's cost to be the expected number of training runs they conduct (over randomness of the protocol execution).[7] We are now ready to state the theorem.

**Theorem 1** (Main Protocol Theorem). *We assume that $f : \{\pm 1\}^N \to [-b, b]$ for some constant $b$. For any $\varepsilon, \delta \in (0, 1)$, Algorithm 1 is a $(\varepsilon, \delta)$-PAC verifier for $\Phi$ (as defined above, with $\mathrm{err}$ measuring the MSE gap). The Verifier's cost function satisfies $\kappa \in O(\log(1/\delta)/\varepsilon^2)$. Furthermore, the interactive protocol requires only two messages.*

### Scalability to Multiple Attribution Tasks.

While our core protocol (Algorithm 1) is presented for verifying attributions with respect to a single model output function $f$, a natural question arises regarding its applicability when attributions are needed for multiple scenarios simultaneously—for instance, across $Z$ different test points or for $Z$ distinct output metrics. Our framework extends efficiently to such cases. The key observation is that, to maintain an overall $(\varepsilon, \delta)$-PAC verification guarantee across all $Z$ attribution tasks, a union bound can be applied to the confidence parameters. This means that the number of challenge subsets requested from the Prover for residual estimation, and the number of local samples used by the Verifier for its final MSE checks, would increase logarithmically in $Z$. We refer to section E for details.

## 3 A "Simple" Non-Interactive Verification Protocol

This section will use standard notation from Boolean Harmonic Analysis. We refer to section A.1 for the necessary definitions (see instead O'Donnell (2014) for the comprehensive treatment).

Before presenting our main interactive protocol, we first outline a simpler, *non-interactive* approach to PAC-verifying empirical influence attribution $\Phi(S)$. This serves to establish a baseline Verifier complexity and motivate the introduction of interaction to achieve greater efficiency.

Recall the verification goal defined in Definition 1. The Verifier V receives a candidate attribution vector $\mathbf{a}'$ from the Prover P and needs to determine if it is $\varepsilon$-close to $\Phi(S)$. Using our chosen error metric, this means verifying if:

$$\mathrm{err}(\mathbf{a}', \Phi(S)) = \mathrm{mse}^{(p)}(f, \langle \mathbf{a}', \cdot \rangle) - \mathrm{mse}^{(p)}(f, \langle \Phi(S), \cdot \rangle) \le \varepsilon.$$

A close relationship exists between $\mathrm{mse}^{(p)}(f, \langle \Phi(S), \cdot \rangle)$ and the *Boolean Fourier coefficients* of $f$ (see section A.1 for the definition and background on Fourier coefficients). In fact, as demonstrated by Saunshi et al. (2022), the optimal linear predictor $\langle \Phi(S), \cdot \rangle$ satisfies the identity:

$$\mathrm{mse}^{(p)}(f, \langle \Phi(S), \cdot \rangle) = \sum_{S \subseteq [N]: |S| \ge 2} \hat{f}_S^2 \triangleq B_{\ge 2}.$$

Therefore, the verification condition is equivalent to checking if the Prover's solution $\mathbf{a}'$ satisfies:

$$\mathrm{mse}^{(p)}(f, \langle \mathbf{a}', \cdot \rangle) \le B_{\ge 2} + \varepsilon.$$

The "residual estimation" algorithm developed by Saunshi et al. (2022) provides a useful tool. As stated in Theorem 2, their algorithm, denoted RESIDUALESTIMATION, allows estimating $B_{\ge 2}$ up

---

[7]This is but one way to define cost. For now, we only mention that this notion of *sample complexity* in data attribution is a commonly used efficiency metric in the field (see e.g, Ilyas et al. (2022); Park et al. (2023)). We emphasize this metric as each retraining constitutes a significant computational effort, potentially scaling with the dataset size $N$ itself. Hence, it captures the primary bottleneck for the Verifier.

to an additive error $\varepsilon'$ with high probability, using only $n = O(1/(\varepsilon')^3 \cdot \mathrm{polylog}(1/\delta))$ evaluations (samples) of the function $f$. Importantly, this algorithm estimates the residual error *without* needing to compute the optimal linear predictor $\Phi(S)$ itself.

**Theorem 2** (Residual Estimation, restated (Saunshi et al., 2022, Theorem 3.2)). *Let $f : \{\pm 1\}^N \to \mathbb{R}$. Let $\hat{B}_{\geq 2}$ be the output of the residual estimation algorithm RESIDUALESTIMATION using degree $d = 2$ polynomial fitting based on noise stability estimates at points $[0, \rho, 2\rho]$ (along with $\rho = 1$), obtained using a total budget of $n$ calls to $f$. If $n = O(1/\varepsilon^3)$ and $\rho = \Theta(\sqrt{\varepsilon})$, then with high probability (e.g., $1 - \delta$ for small constant $\delta$),*

$$|\hat{B}_{\geq 2} - B_{\geq 2}| < \varepsilon,$$

*where $B_{\geq 2} = \sum_{S:|S|\geq 2} \hat{f}_S^2$ is the true residual error under the $\mathcal{B}_p$ distribution.*

This immediately suggests a non-interactive PAC-verification protocol:

1. **Prover P:** Computes $\mathbf{a}'$ (e.g., its best estimate of $\Phi(S)$) and sends it to V.

2. **Verifier V (Local Computation):**
   (a) Estimates the MSE of the Prover's solution: Compute $\widehat{\mathrm{mse}} \approx \mathrm{mse}^{(p)}(f, \langle \mathbf{a}', \cdot \rangle)$ by drawing $M = O(1/\varepsilon^2 \cdot \mathrm{polylog}(1/\delta))$ samples $x \sim \mathcal{B}_p$, locally training models to evaluate $f(x)$ for each sample, and calculating the empirical average squared error $(f(x) - \langle \mathbf{a}', x \rangle)^2$.
   (b) Estimates the optimal residual error: Compute $\hat{B}_{\geq 2} \approx B_{\geq 2}$ by running RESIDUALESTIMATION$(f, \rho, n)$ locally, setting the budget $n = O(1/\varepsilon^3 \cdot \mathrm{polylog}(1/\delta))$ and $\rho = \Theta(\sqrt{\varepsilon})$. This requires locally evaluating $f$ on $n$ (pairs of) input subsets.

3. **Verifier V (Decision):** If $\widehat{\mathrm{mse}} \leq \hat{B}_{\geq 2} + \varepsilon''$, output $\mathbf{a}'$ (where $\varepsilon''$ accounts for the combined estimation errors from steps (a) and (b), we can assume $\varepsilon'' \approx \varepsilon$). Otherwise, output $\mathrm{abort}$.

See section B in the appendix for an overview of how the residual estimation algorithm works.

**Informal Analysis of the Non-Interactive Protocol.** Completeness and Soundness of this protocol can be seen to follow from the accuracy guarantees of standard MSE estimation (via Hoeffding/Chernoff bounds) and the guarantee of the residual estimation algorithm (Theorem 2), using a union bound over the failure probabilities.

The Verifier's efficiency is measured by the number of calls to $f$ (i.e., model trainings). Step (a) requires $O(1/\varepsilon^2 \cdot \mathrm{polylog}(1/\delta))$ calls. Step (b), the residual estimation, requires $O(1/\varepsilon^3 \cdot \mathrm{polylog}(1/\delta))$ calls. The overall Verifier cost $\kappa(\varepsilon, \delta)$ is therefore dominated by the residual estimation:

$$\kappa_{\mathrm{residual}}(\varepsilon, \delta) \in O(1/\varepsilon^3 \cdot \mathrm{polylog}(1/\delta)).$$

While this cost is independent of the dataset size $N$, the cubic dependence on $1/\varepsilon$ can still be substantial.

**Motivation for Interaction.** The $O(1/\varepsilon^3)$ complexity arises directly from the need to execute the residual estimation algorithm locally. This motivates exploring an *interactive* approach. If V could somehow leverage the computational power of P to perform the $O(1/\varepsilon^3)$ function evaluations required for residual estimation, while still retaining statistical guarantees against a potentially dishonest P, the Verifier's workload could potentially be reduced.

This is precisely the approach taken in the remainder of this paper. We develop a two-message interactive protocol where V challenges P to perform the function evaluations needed for residual estimation. To ensure P's responses are trustworthy, V employs a *spot-checking* mechanism, verifying a small random subset of P's computations locally. As we will show, this spot-checking strategy, combined with a proof that the residual estimation algorithm is robust to a limited number of undetected errors (Section D.1, Lemma 1), allows V to achieve PAC verification with a reduced cost of $O(1/\varepsilon^2 \cdot \mathrm{polylog}(1/\delta))$, matching the complexity of the simpler MSE estimation step.

## 4 An Improved Interactive Protocol

In this section, we describe our improved interactive protocol. The formal protocol and its analysis are given in Appendix section C. Our protocol uses as a subroutine the residual estimation algorithm

(Algorithm 1) of Saunshi et al. (2022). As part of the analysis of our improved protocol, we will need to analyze this residual estimation algorithm precisely, using the details of the algorithm's implementation. In Appendix section B, we give a detailed overview of the algorithm.

- **Round 1 (V → P): Challenge Setup.**
  - The Verifier V generates a set of computational "challenges" for the Prover P. These challenges consist of $|E| = O(\log(1/\delta)/\varepsilon^3)$ randomly selected subsets of the training data $S \sim \mathcal{B}_p$, using specific random seeds provided by V. Here $|E|$ is the number of samples that will be used for the residual estimation algorithm.
  - V secretly designates a randomly chosen subset of $k = O(\log(1/\delta)/\varepsilon^2)$ of these challenges for later "spot-checking."
  - V sends the list of training subsets and their corresponding random seeds to P.
- **Round 2 (P → V): Honest Prover's Response.**
  - The Prover P computes the optimal data attribution scores for the full dataset $S$.
  - P performs the training runs requested by V in Round 1, using the specified subsets and random seeds.
  - P sends the resulting trained model weights for all challenges, along with the computed attribution scores $\mathbf{a}$, back to V.
- **Round 3 (V): Verification.**
  - **Spot-Checking:** V repeats training runs for the $k$ challenges designated for spot-checking in Round 1. V checks if the resulting models are equivalent to those sent by P (using an equivalence checking procedure). If any inequivalence occurs, V aborts.
  - **Consistency Check:** If the spot-checks pass, V uses the model results provided by P (for all non-spot-checked challenges) to perform two estimations:
    (a) For $O(\log(1/\delta)/\varepsilon^2)$ models, evaluate the model output function $f$, and use them to estimate $\alpha = \mathrm{mse}^{(p)}(f, \langle \mathbf{a}, \cdot \rangle)$.
    (b) Run the residual estimation algorithm (Theorem 2) to estimate $\beta = \min_{\mathbf{w}} \mathrm{mse}^{(p)}(f, \langle \mathbf{w}, \cdot \rangle)$.
  - V compares $\alpha$ and $\beta$. If $\alpha$ and $\beta$ deviate by more than $\varepsilon$, V aborts.
  - **Output:** If all checks pass, V accepts and outputs the attribution scores provided by P. Otherwise, V outputs $\mathrm{abort}$.

Now that we have a sense of the key steps involved in the improved protocol, we can sketch a proof of its guarantees.

## 4.1 Proof Sketch of Main Theorem

Due to space constraints we defer the full proof to Appendix section D, and simply sketch the main ideas behind each correctness guarantee for the protocol.

**Completeness.** To argue completeness, we start with the observation that if the Prover is honest, then the "spot-checks" used in the protocol will pass. In this case, the Verifier is able to accurately estimate both the optimal linear prediction error using the residual estimation algorithm, and the actual error of the Prover's candidate attribution scores using local training runs.

Since the honest Prover submits the optimal scores $\mathbf{a}^\star = \Phi(S)$ (or $\epsilon$-close to optimal), these two errors are close, causing the final consistency check to pass and the Verifier to accept the correct attribution.

**Soundness.** To prove soundness, we will consider different cases. First, where a dishonest Prover lies about "many" of the challenges, and second, where the dishonest Prover lies about "a few" of the challenges. In case one, we show that the Prover will be caught with high probability by the Verifier's random spot-checks; intuitively if the Prover lies about more than a certain fraction of the requested trainings, the Verifier will catch the Prover after conducting the appropriate amount of spot checks, with high probability. In this case, the Verifier outputs "abort" with high probability. In case two, the Prover might lie about less than that fraction of the challenges, and we will show that the Verifier's spot checks may all pass, but that despite this, the residual estimation will still be good. At a technical level, in Lemma 1 we show that the residual estimation algorithm of Saunshi et al. (2022) is robust to $O(1/\epsilon)$ adversarial corruptions. As a result, in this case, even if the Prover has submitted incorrect

candidate attribution scores $\mathbf{a}'$, then the Verifier's local estimate of the high error $\mathrm{mse}^{(p)}(f, \langle \mathbf{a}', \cdot \rangle)$ will then significantly exceed the estimated optimal error, causing the final consistency check to fail and the Verifier to abort.

**Efficiency.** To show efficiency, we will directly estimate the expected number of model trainings done by the Verifier, which is straightforward: it is the sum of the number of model trainings in Round 3, which consists of $O(\log(1/\delta)/\epsilon^2)$ spot checks, and $O(\log(1/\delta)/\epsilon^2)$ to estimate the MSE. Thus the overall cost is $O(\log(1/\delta)/\epsilon^2)$.

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

# A Preliminaries

## A.1 Harmonic Analysis on the p-biased Cube

Our work (and important prior work like Goldwasser et al. (2021) and Saunshi et al. (2022)) leverage tools that are grounded in the analysis of real-valued functions defined on the discrete hypercube, specifically under a non-uniform measure known as the $p$-biased distribution (see e.g., O'Donnell (2014) for a comprehensive overview of Boolean Fourier analysis). In this section, we will go over the necessary concepts for understanding the mathematics presented in this paper.

Let $N = |S|$ be the size of the training dataset. Recall that we represent a subset of the training data by a vector $x \in \{\pm 1\}^N$, where $x_i = 1$ indicates the $i$-th datapoint is included and $x_i = -1$ indicates it is excluded. The model's behavior (e.g., loss or logit difference on a test point) after training on the subset represented by $x$ is given by a function $f : \{\pm 1\}^N \to \mathbb{R}$.

We consider the $p$-biased distribution $\mathcal{B}_p$ over $\{\pm 1\}^N$, where each coordinate $x_i$ is independently chosen to be 1 with probability $p$ and $-1$ with probability $1-p$. Let $\mu = \mathbb{E}[x_i] = p - (1-p) = 2p - 1$ and $\sigma^2 = \mathrm{Var}(x_i) = \mathbb{E}[x_i^2] - (\mathbb{E}[x_i])^2 = 1 - (2p-1)^2 = 4p(1-p)$. The uniform distribution corresponds to $p = 1/2$, where $\mu = 0$ and $\sigma^2 = 1$.

The space of functions $f : \{\pm 1\}^N \to \mathbb{R}$ forms a vector space equipped with the inner product $\langle f, g \rangle_{\mathcal{B}_p} = \mathbb{E}_{x \sim \mathcal{B}_p}[f(x)g(x)]$. An orthonormal basis for this space is given by the characters $\{\phi_S\}_{S \subseteq [N]}$, defined as:

$$\phi_S(x) = \prod_{i \in S} \frac{x_i - \mu}{\sigma}$$

These basis functions satisfy $\mathbb{E}[\phi_S(x)] = 0$ for $S \neq \emptyset$ and $\langle \phi_S, \phi_T \rangle_{\mathcal{B}_p} = \delta_{S,T}$ (Kronecker delta). Any function $f$ can be uniquely expanded in this basis as:

$$f(x) = \sum_{S \subseteq [N]} \hat{f}_S \phi_S(x),$$

where $\hat{f}_S = \langle f, \phi_S \rangle_{\mathcal{B}_p} = \mathbb{E}_{x \sim \mathcal{B}_p}[f(x)\phi_S(x)]$ are the Fourier coefficients of $f$. Parseval's identity states:

$$\|f\|_{\mathcal{B}_p}^2 = \mathbb{E}_{x \sim \mathcal{B}_p}[f(x)^2] = \sum_{S \subseteq [N]} \hat{f}_S^2$$

The coefficients $\hat{f}_S$ capture the contribution of interactions among the datapoints in $S$ to the function $f$. Of particular importance are the degree-1 coefficients $\hat{f}_{\{i\}}$, which, up to scaling by $2/\sigma$, correspond to the average influence of datapoint $i$ and the optimal coefficients for linear datamodels (Saunshi et al., 2022, Theorem 2.2). To denote the total Fourier weight at degree $k$, let

$$B_k \triangleq \sum_{S : |S| = k} \hat{f}_S^2 \tag{3}$$

To analyze the structure of $f$, we use the concept of noise stability.

**Definition 2** ($\rho$-correlated variables). *For $x \in \{\pm 1\}^N$ drawn from $\mathcal{B}_p$, we say a random variable $x'$ is $\rho$-correlated to $x$ if it is sampled coordinate-wise independently as follows: For each $i \in [N]$,*

- *If $x_i = 1$, then $x_i' = -1$ with probability $(1-p)(1-\rho)$, and $x_i' = 1$ otherwise.*

- *If $x_i = -1$, then $x_i' = 1$ with probability $p(1-\rho)$, and $x_i' = -1$ otherwise.*

*Crucially, $x'$ is also distributed according to $\mathcal{B}_p$.*

The *noise stability* of $f$ at noise rate $\rho \in [0, 1]$ is defined as $h_f(\rho) = \mathbb{E}_{x,x'}[f(x)f(x')]$, where $x \sim \mathcal{B}_p$ and $x'$ is $\rho$-correlated to $x$. It admits a simple expression in terms of the Fourier weights:

$$h_f(\rho) = \sum_{k=0}^{N} B_k \rho^k \tag{4}$$

This shows that the noise stability is a polynomial in $\rho$, whose coefficients are precisely the total Fourier weights $B_k$ at each degree $k$.

# B  Efficient Estimation of Linear Datamodel Residual

As established by Saunshi et al. (2022) (specifically their Theorem 2.2), the quality of the best linear datamodel approximation $\theta_0 + \sum_{i=1}^{N} \theta_i x_i$ for a function $f : \{\pm 1\}^N \to \mathbb{R}$ under the $p$-biased distribution $\mathcal{B}_p$ is determined by its residual error:

$$R(\theta^\star) \triangleq B_{\geq 2} = \sum_{S \subseteq [N]:|S| \geq 2} \hat{f}_S^2$$

where $\theta^\star$ denotes the coefficients that minimize MSE of the datamodel $\theta_0 + \sum_{i=1}^{N} \theta_i x_i$ with respect to the model output function $f$ (Saunshi et al. (2022) use the notation $R(\theta^\star)$, so we include that here for clarity, even though throughout the paper we refer to MSE explicitly).

Saunshi et al. (2022) cleverly leverage the connection between degree-$k$ weight of $f$ and the noise stability of $f$, $h_f(\rho)$. Recall from (4) that

$$h_f(\rho) = \sum_{k=0}^{N} B_k \rho^k$$

Separately, the total squared norm is $B = \|f\|_{\mathcal{B}_p}^2 = \sum_{k=0}^{N} B_k = h_f(1)$. Therefore, the residual error can be expressed purely in terms of the degree-0 and degree-1 Fourier weights and the total norm:

$$B_{\geq 2} = B - B_0 - B_1 = h_f(1) - B_0 - B_1$$

The algorithm from Saunshi et al. (2022) leverages this fact as follows:

1. **Estimate Noise Stability at Key Points:** Choose a small number $k$ of distinct noise levels $\{\rho_1, \ldots, \rho_k\}$ (e.g., $0, \rho, 2\rho$ for some small $\rho$) and additionally use $\rho = 1$. For each chosen $\rho_j$ (and for $\rho = 1$), estimate the noise stability value $h_f(\rho_j)$ by sampling multiple pairs $(x, x')$ of $\rho_j$-correlated inputs drawn from $\mathcal{B}_p$ and averaging the product $f(x)f(x')$. Let these estimates be $\hat{y}_j \approx h_f(\rho_j)$ and $\hat{B} \approx h_f(1)$.

2. **Low-Degree Polynomial Fitting:** Since $h_f(\rho) = B_0 + B_1\rho + B_2\rho^2 + \ldots$, use the estimated values $(\rho_j, \hat{y}_j)$ from the previous step to fit a low-degree polynomial (e.g., degree $d = 2$) $P(\rho) = \sum_{i=0}^{d} \hat{B}_i \rho^i$. This is done via (non-negative) least squares, solving for the coefficients $\hat{B}_0, \hat{B}_1, \ldots, \hat{B}_d$.

3. **Calculate Residual Estimate:** Combine the estimated total norm $\hat{B} \approx h_f(1)$ and the estimated low-degree coefficients $\hat{B}_0, \hat{B}_1$ obtained from the polynomial fit to compute the final estimate of the residual error: $\hat{B}_{\geq 2} = \hat{B} - \hat{B}_0 - \hat{B}_1$.

This approach allows estimating the quality of the best linear fit using a number of samples that is proportional to $1/\varepsilon^3$ for a desired approximation error $\varepsilon$ but is *independent* of the dimension $N$.

## C  Our Full Protocol

We present our protocol for PAC-verification of empirical influence (Theorem 1), detailed in Algorithm 1.

---

**Algorithm 1** Interactive PAC-Verification Protocol for Empirical Influence ($\Phi$)

---

    **Shared Information:**

        Model Architecture, Training set $S \subseteq \mathcal{X}$ ($|S| = N$), subsampling probability $p$,

        Training loss function $\mathcal{L}$, Stopping criteria for training.

        Model output function $f : \{\pm1\}^N \to \mathbb{R}$ derived from training.

        Equivalence check procedure $\text{CHECKEQUIV}(\theta_1, \theta_2)$.

    **Verifier (V) Input:**

        Approximation parameter $\varepsilon \in (0, 1)$, Confidence parameter $\delta \in (0, 1)$.

    **Round 1: Verifier (V) $\to$ Prover (P)**

1: Sample $M \leftarrow \{x^{(m)}\}_{m=1}^{O(1/\varepsilon^2 \cdot \log(1/\delta))}$ where each $x^{(m)} \sim \mathcal{B}_p$.      $\star$ *Subsets for final MSE check*

2: Choose $\rho \leftarrow \Theta(\sqrt{\varepsilon})$. Sample $n = O(1/\varepsilon^3 \cdot \log(1/\delta))$ pairs $\{(x^{(e)}, x'^{(e)})\}_{e=1}^n$ where $x^{(e)} \sim \mathcal{B}_p$ and $x'^{(e)}$ is $\rho$-correlated to $x^{(e)}$. Let $E$ be the set of all unique subsets appearing in these $n$ pairs (at most $2n$ subsets).
                                                        $\star$ *Subsets for residual estimation*

3: Choose random seeds $R \leftarrow \{r_e\}_{x_e \in E}$ for training models on subsets in $E$.

4: Set spot-check set size $k \leftarrow O(1/\varepsilon^2 \cdot \log(1/\delta))$. Sample spot-check set $C \subseteq E$ of size $k$ uniformly at random.      $\star$ *$k$ will ensure high prob. detection*

5: Send $(E, R)$ to P.      $\star$ *Seeds $R$ are indexed corresponding to subsets in $E$*

    **Round 2: Prover (P) $\to$ Verifier (V)**

6: Compute attribution $\mathbf{a}^\star \leftarrow \Phi(S)$.      $\star$ *Computationally expensive*

7: **For** each $x_e \in E$ **do**

8:     Train model with seed $r_e$ on subset $x_e$ to get weights $\theta_e$.

9: **End For**

10: Let $\boldsymbol{\theta} = \{\theta_e\}_{x_e \in E}$.

11: Send $(\boldsymbol{\theta}, \mathbf{a}^\star)$ to V.      $\star$ *Sends all trained models and the computed attribution*

    **Round 3: Verifier (V) Verification and Output**

12: **Spot-check:**

13: **For** each $x_c \in C$ **do**

14:     Train model with seed $r_c$ on subset $x_c$ to get local weights $\hat{\theta}_c$.      $\star$ V *performs limited training*

15:     Let $\theta'_c$ be the model weights received from a (possibly dishonest) Prover P$'$ for subset $x_c$.

16:     **If not** $\text{CHECKEQUIV}(\hat{\theta}_c, \theta'_c)$ **then**      $\star$ *Check if Prover's result matches Verifier's*

17:         **Output:** abort and **Terminate**.

18:     **End If**

19: **End For**

20:                                                             $\star$ *All spot-checks passed*

21: Use received models $\boldsymbol{\theta}'$ to define the function values $f'(x_e)$ for $x_e \in E$.

22: Compute residual estimate $\hat{B}_{\geq 2} \leftarrow \text{RESIDUALESTIMATION}(f')$ on $E, \rho$, budget $|E|$.

23:                                   $\star$ *Uses function values derived from* P*'s models (for $x_e \notin C$)*

24: **For** each $x_m \in M$ **do**

25:     Train model with fresh random seed $r'_m$ on subset $x_m$ to get weights $\theta_m$.

26:     Evaluate $f(x_m)$ using $\theta_m$.      $\star$ V *trains models for MSE estimate*

27: **End For**

28: Let $g(x) = \langle \mathbf{a}^\star, x \rangle$. Estimate $\widehat{\text{mse}}_p(f, g) \leftarrow \frac{1}{|M|} \sum_{x_m \in M} (f(x_m) - g(x_m))^2$.

29: **If** $\widehat{\text{mse}}_p(f, g) > \hat{B}_{\geq 2} + \varepsilon/2$ **then**

30:     **Output:** abort and **Terminate**.

31: **Else**

32:     **Output: $\mathbf{a}^\star$**      $\star$ *Accept the Prover's attribution*

33: **End If**

---

**The Equivalence Checking Procedure.** Note that we assume access to an equivalence-checking procedure between model training runs. The exact implementation of CHECKEQUIV depends on the training setup. Ideally, even with fully deterministic training (with a fixed architecture, data subset $x_c$, and randomness seed $r_c$), this check could be a direct comparison of model weights ($\theta_c = \theta'_c$). However, practical ML training can exhibit non-determinism (e.g., due to GPU parallelism). In such cases, CHECKEQUIV might involve comparing model outputs on a held-out test set, comparing final loss values within a tolerance, or potentially using cryptographic commitments or hashes if sufficient determinism can be enforced.

# D  Proof of Main Theorem

In this section, we will prove that the protocol outlined in Algorithm 1 works to witness Theorem 1. For now, let us start with an outline of our proof.

**Completeness.** To argue completeness, we start with the observation that if the Prover is honest, then the "spot-checks" used in the protocol will pass. In this case, the Verifier is able to accurately estimate both the optimal linear prediction error ($B_{\geq 2}$) using the residual estimation algorithm, and the actual error of the Prover's candidate attribution scores using local training runs.

Since the honest Prover submits the optimal, or near optimal, scores $\mathbf{a}^\star = \Phi(S)$, these two errors are close, causing the final consistency check to pass and the Verifier to accept the correct attribution.

**Soundness.** To prove soundness, we will consider different cases. First, where a dishonest Prover lies about "many" of the requested model trainings ($\boldsymbol{\theta}'$), and second, where the dishonest Prover lies on "a few" of the requested model trainings.

In case one, we show that the Prover will be caught with high probability by the Verifier's random spot-checks. In this case, the Verifier outputs "abort" with high probability.

In case two, we will show that the Verifier's spot checks may all pass, but that despite this, the residual estimation will still be good. That is, we show that it is robust to some adversarial corruptions. We prove a standalone lemma to accomplish this.

As a result, in this case, even if the Prover has submitted incorrect candidate attribution scores $\mathbf{a}'$, then the Verifier's local estimate of the high error $\mathrm{mse}^{(p)}(f, \langle \mathbf{a}', \cdot \rangle)$ will then significantly exceed the estimated optimal error $\hat{B}_{\geq 2}$, causing the final consistency check to fail and the Verifier to abort.

**Efficiency.** To show efficiency, we will directly estimate the expected number of model trainings done by the Verifier.

## D.1  Supporting Lemma: Robust Residual Estimation

Before proving the main theorem, we establish the robustness of the RESIDUALESTIMATION subroutine against a limited number of adversarial corruptions.

**Lemma 1** (Robust Residual Estimation). *Let $f : \{\pm 1\}^N \to [-b, b]$ be bounded. Let $\hat{B}_{\geq 2}$ be the output of* RESIDUALESTIMATION$(f, \rho, n)$ *using degree $d = 2$, points $[0, \rho, 2\rho]$ (and $\rho = 1$), and based on $n$ evaluations of $f$ run on the same sample points, but where an adversary corrupts up to $m = O(1/\varepsilon)$ of the $n$ function evaluations, by replacing $f(x)$ with $f'(x) \in [-b, b]$. Let $n = O(1/\varepsilon^3 \log(1/\delta))$, and $\rho = \Theta(\sqrt{\varepsilon})$.*

*Then, with probability at least $1 - \delta/4$ (over the internal randomness of the algorithm):*

$$|\hat{B}_{\geq 2} - B_{\geq 2}| = \tilde{O}(b^2 \varepsilon)$$

*where the constant in $\tilde{O}(\cdot)$ depends on $\delta$ but not on $n, m, \varepsilon, N$.*

*Proof of Lemma 1.* Let us first recap the RESIDUALESTIMATION algorithm.

RESIDUALESTIMATION uses $n$ total function evaluations, distributed among estimating $h_f(0)$, $h_f(\rho)$, $h_f(2\rho)$ and $h_f(1)$. Let $n_0, n_\rho, n_{2\rho}$ be the number of pairs $(x, x')$ sampled to estimate $h_f(0), h_f(\rho), h_f(2\rho)$ respectively, and let $N_1$ be the number of samples $x$ used to estimate $h_f(1) = \mathbb{E}\left[f(x)^2\right]$. The total number of function evaluations is $n = (n_0 + n_\rho + n_{2\rho}) \times 2 + N_1$.

Let $\tilde{y}_0, \tilde{y}_\rho, \tilde{y}_{2\rho}$ be the empirical estimates of $h_f(0), h_f(\rho), h_f(2\rho)$ obtained using the original function $f$ evaluations from the $n$ samples. Let $\tilde{B}$ be the estimate of $h_f(1) = \mathbb{E}\left[f(x)^2\right]$. Let $\tilde{\mathbf{y}} = [\tilde{y}_0, \tilde{y}_\rho, \tilde{y}_{2\rho}]^T$. The algorithm RESIDUALESTIMATION uses these estimates, which are not corrupted by any adversary, to compute the uncorrupted residual estimate $\tilde{B}_{\geq 2}$. Specifically, a vector $\tilde{\mathbf{z}} = [\tilde{B}_0, \tilde{B}_1, \tilde{B}_2]^T$ is found by solving the constrained least squares problem

$$\arg\min_{\mathbf{z} \geq 0} \|\mathbf{A}\mathbf{z} - \tilde{\mathbf{y}}\|_2^2 \tag{5}$$

Here,

$$\mathbf{A} = \begin{pmatrix} 1 & 0 & 0 \\ 1 & \rho & \rho^2 \\ 1 & 2\rho & 4\rho^2 \end{pmatrix} \tag{6}$$

Then $\tilde{B}_{\geq 2} = \tilde{B} - \tilde{z}_1 - \tilde{z}_2$ ($\tilde{z}_1$ would correspond to $B_0$, $\tilde{z}_2$ would correspond to $B_1$).

Now, we consider the setting where an adversary corrupts $m = O(1/\varepsilon)$ of the $n$ function evaluations. Let $f'(x)$ denote the value used in the computation after potential corruption. Since $f : \{\pm 1\}^N \to [-b, b]$, we also have $f'(x) \in [-b, b]$. Let $\hat{y}_0, \hat{y}_\rho, \hat{y}_{2\rho}$ be the estimates using the potentially corrupted values $f'$, and let $\hat{B}$ be the estimate of $\mathbb{E}\left[f'(x)^2\right]$. Let $\hat{\mathbf{y}} = [\hat{y}_0, \hat{y}_\rho, \hat{y}_{2\rho}]^T$. Let:

$$\hat{\mathbf{z}} = [\hat{B}_0, \hat{B}_1, \hat{B}_2]^T = \arg\min_{\mathbf{z} \geq 0} \|\mathbf{A}\mathbf{z} - \hat{\mathbf{y}}\|_2^2 \tag{7}$$

and $\hat{B}_{\geq 2} = \hat{B} - \hat{z}_1 - \hat{z}_2$.

Our goal is to bound $|\hat{B}_{\geq 2} - B_{\geq 2}|$, where $B_{\geq 2}$ is the "true value" (assume exact estimation). We use the triangle inequality:

$$|\hat{B}_{\geq 2} - B_{\geq 2}| \leq |\hat{B} - B| + |\hat{z}_1 - z_1| + |\hat{z}_2 - z_2|$$

Here $z_1, z_2$ are the "true coefficients" of the noise stability polynomial (recall equation 4).

To do this, we will apply the triangle inequality again, and bound each of the terms on the RHS above, by $|\hat{B} - B| \leq |\hat{B} - \tilde{B}| + |\tilde{B} - B|$, and similarly, $|\hat{z}_i - z_i| \leq |\hat{z}_i - \tilde{z}_i| + |\tilde{z}_i - z_i|$.

First, bound the difference in the estimates $\hat{y}_i - \tilde{y}_i$ and $\hat{B} - \tilde{B}$. Consider $\hat{y}_\rho - \tilde{y}_\rho$. This is the difference between the average of $f'(x)f'(x')$ and $f(x)f(x')$ over $n_\rho = \Theta(n)$ pairs. An adversary corrupts $m$ function evaluations total. Each term $f(x)f(x')$ uses two evaluations. At most $m$ terms in the average can be affected by corruption. Let $S_\rho \subseteq [n_\rho]$ be the indices of the affected terms. $|S_\rho| \leq m$.

$$\hat{y}_\rho - \tilde{y}_\rho = \frac{1}{n_\rho} \sum_{j \in S_\rho} (f'(x_j)f'(x_j') - f(x_j)f(x_j'))$$

Since $|f(x)f(x')| \leq b^2$ and $|f'(x)f'(x')| \leq b^2$, the difference in each term is bounded by $2b^2$.

$$|\hat{y}_\rho - \tilde{y}_\rho| \leq \frac{1}{n_\rho} \sum_{j \in S_\rho} 2b^2 \leq \frac{m \cdot 2b^2}{n_\rho}$$

Similarly,

$$|\hat{y}_0 - \tilde{y}_0| \leq \frac{m \cdot 2b^2}{n_0} \tag{8}$$

and

$$|\hat{y}_{2\rho} - \tilde{y}_{2\rho}| \leq \frac{m \cdot 2b^2}{n_{2\rho}} \tag{9}$$

Also,

$$|\hat{B} - \tilde{B}| = |\frac{1}{N_1} \sum_{j \in S_1} (f'(x_j)^2 - f(x_j)^2)| \leq \frac{m \cdot 2b^2}{N_1} \tag{10}$$

Then the maximum element-wise error is bounded:

$$\|\hat{\mathbf{y}} - \tilde{\mathbf{y}}\|_\infty \leq \max\left(\frac{2mb^2}{n_0}, \frac{2mb^2}{n_\rho}, \frac{2mb^2}{n_{2\rho}}\right) = O\left(\frac{mb^2}{n}\right)$$

Since $n = O(1/\varepsilon^3 \log(1/\delta))$ and $m = O(1/\varepsilon)$, we have

$$\|\hat{\mathbf{y}} - \tilde{\mathbf{y}}\|_\infty = O\left(\frac{(1/\varepsilon)b^2}{1/\varepsilon^3}\right) = O(b^2\varepsilon^2)$$

Similarly, $|\hat{B} - \tilde{B}| = O(b^2\varepsilon^2)$.

Now, we can bound the difference between the uncorrupted estimates $\tilde{y}_0, \tilde{y}_\rho, \tilde{y}_{2\rho}$ and the "true values" $h_f(0), h_f(\rho), h_f(2\rho)$ using standard concentration inequalities. For instance, see Lemma A.1 of Saunshi et al. (2022), which shows that the estimations are close enough to the true values with probability $1 - \delta$, i.e. $\tilde{y}_\rho = \tilde{h}(\rho) = h(\rho) + \gamma_\rho$ where $|\gamma_\rho| \leq \gamma$ where $\gamma = O(\sqrt{\log(1/\delta)/n})$.

At this point, we can combine the adversarial error bound and estimation error bounds for the noise sensitivity estimates, and complete the analysis as do Saunshi et al. (2022) in their analysis of RESIDUALESTIMATION.

From above, we have, with high probability,

$$\|\hat{\mathbf{y}} - \mathbf{y}\|_\infty \leq \|\hat{\mathbf{y}} - \tilde{\mathbf{y}}\|_\infty + \|\tilde{\mathbf{y}} - \mathbf{y}\|_\infty \leq O(b^2\varepsilon^2) + O(\varepsilon^{3/2}) \leq O(b^2\varepsilon^{3/2})$$

Then, applying the analysis from Saunshi et al. (2022) (Theorem 3.2), of how the perturbed estimates affect the constrained least squares solutions, we get

$$|\hat{B} - B| \text{ and } |\hat{z}_1 - z_1| \text{ and } |\hat{z}_2 - z_2| \leq O\left(\frac{b^2\varepsilon^{3/2}}{\rho} + B_{\geq 3}\rho^2\right)$$

Hence,

$$|\hat{B}_{\geq 2} - B_{\geq 2}| \leq O\left(\frac{b^2\varepsilon^{3/2}}{\rho} + B_{\geq 3}\rho^2\right)$$

Finally, for the setting of $\rho = \sqrt{\varepsilon}$, and observing that $B_{\geq 3} \leq b^2$ we obtain the desired expression.

$$|\hat{B}_{\geq 2} - B_{\geq 2}| \leq O(b^2\varepsilon)$$

$\square$

## D.2   Proof of Main Theorem

*Proof of Theorem 1.* We prove Theorem 1 by establishing Completeness, Soundness, and Efficiency as per Definition 1. Recall that we assume $|f(x)| \leq b = O(1)$. We set internal protocol confidence parameters for subroutines to $\delta' = \delta/4$.

## D.3   Completeness

For completeness we can assume both $\mathsf{V}$ and $\mathsf{P}$ follow the protocol.

1. **Prover's Output:** Honest $\mathsf{P}$ computes $\mathbf{a}^\star = \Phi(S)$ and correct models $\boldsymbol{\theta}$. We have $\mathrm{err}(\mathbf{a}^\star, \Phi(S)) = 0 \leq \varepsilon$. $\mathsf{P}$ sends $(\boldsymbol{\theta}, \mathbf{a}^\star)$.

2. **Verifier's Spot-Check:** Passes with probability 1, assuming deterministic training given fixed random seeds.

3. **Verifier's Consistency Check:**

   (a) *Residual Estimation:* $\mathsf{V}$ uses correct $\boldsymbol{\theta}$ for $f$ values. By Theorem 2, using $|E| = O(1/\varepsilon^3 \log(1/\delta'))$, $|\hat{B}_{\geq 2} - B_{\geq 2}| < \varepsilon/4$ with probability at least $1 - \delta'$.

   (b) *MSE Estimation:* $\mathsf{V}$ uses local trainings for $M$. With $|M| = O(1/\varepsilon^2 \log(1/\delta'))$, by Hoeffding's inequality, $|\widehat{\mathrm{mse}}_p(f, \langle \mathbf{a}^\star, \cdot \rangle) - B_{\geq 2}| < \varepsilon/4$ with probability at least $1 - \delta'$.

   (c) *Final Check:* Compare $\widehat{\mathrm{mse}}_p(f, \langle \mathbf{a}^\star, \cdot \rangle)$ with $\hat{B}_{\geq 2} + \varepsilon/2$. With probability at least $1 - 2\delta'$, both estimates are within $\varepsilon/4$ of $B_{\geq 2}$. Then $\widehat{\mathrm{mse}}_p \leq B_{\geq 2} + \varepsilon/4$ and $\hat{B}_{\geq 2} + \varepsilon/2 \geq (B_{\geq 2} - \varepsilon/4) + \varepsilon/2 = B_{\geq 2} + \varepsilon/4$. So, $\widehat{\mathrm{mse}}_p \leq \hat{B}_{\geq 2} + \varepsilon/2$. Therefore the final check passes.

4. **Verifier's Output:** $\mathsf{V}$ outputs $\mathbf{a}^\star$ with $\mathrm{err}(\mathbf{a}^\star, \Phi(S)) = 0 \leq \varepsilon$.

The overall success probability is at least $1 - 2\delta' = 1 - \delta/2 \geq 1 - \delta$. Therefore completeness is proved.

## D.4 Soundness

To analyze soundness, we will proceed by case analysis. The first case deals with a scenario where the Prover is *very* dishonest, in the sense that many of the requested model trainings from the set $E$ (see Round 1 of the protocol) are wrong.

The second case will be mutually exclusive, and deal with the scenario where the Prover is *mildly* dishonest, in the sense that *not too many* of the requested model trainings are wrong. In this second case, the residual estimation robustness lemma (lemma 1) will be useful.

To begin, recall that the dishonest Prover $\mathsf{P}'$ is sending $(\boldsymbol{\theta}', \mathbf{a}')$. Let $W$ be the set of corrupted challenges where $\theta'_e$ is not equivalent to a correctly trained model, and let the number of corruptions be $m = |W|$. Let $|E|$ be the total number of challenges sent to the Prover for residual estimation. We define the threshold for "many" corruptions by setting a critical point $m^\star = c/\varepsilon$ for a sufficiently large constant $c$. The number of spot-checks is $k = O(1/\varepsilon^2 \cdot \log(1/\delta))$.

**Case 1: $m > m^\star$.** In this case, we show that a spot-check will fail with high probability. The Verifier samples a set $C$ of $k$ challenges to check. The Prover is caught if $C \cap W \neq \emptyset$. The probability that the Prover is *not* caught is the probability that all $k$ checks land outside of the corrupted set $W$. This probability can be upper-bounded by sampling with replacement:

$$\Pr[\text{not caught}] \leq \left( \frac{|E| - m}{|E|} \right)^k = \left( 1 - \frac{m}{|E|} \right)^k$$

Since $m > m^\star = c/\varepsilon$, we have:

$$\Pr[\text{not caught}] < \left( 1 - \frac{c/\varepsilon}{|E|} \right)^k \leq e^{-k \cdot (c/\varepsilon)/|E|}$$

Substituting $k = C'/\varepsilon^2 \cdot \log(4/\delta)$ and $|E| = C_E/\varepsilon^3 \cdot \log(1/\delta)$ for constants $C', C_E$, we get the exponent:

$$-\frac{C' \log(4/\delta)}{\varepsilon^2} \frac{c/\varepsilon}{C_E \log(1/\delta)/\varepsilon^3} = -\frac{C'c}{C_E} \frac{\log(4/\delta)}{\log(1/\delta)}$$

By choosing the constant $c$ in the definition of $m^\star$ to be large enough, this exponent can be made smaller than $-\ln(4/\delta)$, ensuring the failure probability is less than $\delta/4$. Thus, we conclude $\mathsf{V}$ aborts with probability at least $1 - \delta/4$.

**Case 2: $m \leq m^\star$.** In this second case, all spot-checks may pass with some probability. Let us (conservatively) assume that all checks indeed pass, and then consider how this affects the Verifier's output.

We analyze the protocol, step by step, in this case.

(1) *Robust Residual Estimation:* $\mathsf{V}$ runs RESIDUALESTIMATION using the function values derived from the models $\boldsymbol{\theta}'$ sent by the Prover. Since $m \leq m^\star = O(1/\varepsilon)$, the number of corruptions is within the required bound for Lemma 1. Therefore, the lemma applies directly. We have $|\hat{B}_{\geq 2} - B_{\geq 2}| < \varepsilon/4$ with probability at least $1 - \delta'$.

(2) *MSE Estimation:* $\mathsf{V}$ estimates $\widehat{\mathrm{mse}}_p(f, g')$ for $g' = \langle \mathbf{a}', \cdot \rangle$. This happens with local samples in the set $M$. Therefore, we can conclude that $|\widehat{\mathrm{mse}}_p(f, g') - \mathrm{mse}^{(p)}(f, g')| < \varepsilon/4$ with probability at least $1 - \delta'$, by applying standard Chernoff bounds.

(3) *Final Check Condition:* Assume $\mathrm{err}(\mathbf{a}', \Phi(S)) > \varepsilon$, which means $\mathrm{mse}^{(p)}(f, g') > B_{\geq 2} + \varepsilon$. with probability at least $1 - 2\delta'$, we have $\widehat{\mathrm{mse}}_p(f, g') > \mathrm{mse}^{(p)}(f, g') - \varepsilon/4 > (B_{\geq 2} + \varepsilon) - \varepsilon/4 = B_{\geq 2} + 3\varepsilon/4$. Also, $\hat{B}_{\geq 2} < B_{\geq 2} + \varepsilon/4$. Then $\hat{B}_{\geq 2} + \varepsilon/2 < (B_{\geq 2} + \varepsilon/4) + \varepsilon/2 = B_{\geq 2} + 3\varepsilon/4$. Since $\widehat{\mathrm{mse}}_p(f, g') > B_{\geq 2} + 3\varepsilon/4$ and $B_{\geq 2} + 3\varepsilon/4 > \hat{B}_{\geq 2} + \varepsilon/2$, the condition $\widehat{\mathrm{mse}}_p(f, g') > \hat{B}_{\geq 2} + \varepsilon/2$ holds.

(4) *Verifier Output:* If $\text{err}(\mathbf{a}', \Phi(S)) > \varepsilon$ and $m \leq m^\star$, V outputs abort with probability at least $1 - 2\delta'$.

**Combining Cases:** If $\text{err}(\mathbf{a}', \Phi(S)) > \varepsilon$, the probability V does **not** abort is at most $\Pr[\text{Case 1 fails}] + \Pr[\text{Case 2 fails}] \leq \delta/4 + 2\delta' = \delta/4 + \delta/2 = 3\delta/4 \leq \delta$. Therefore Soundness holds.

### D.5 Efficiency

Finally, we consider efficiency. The Verifier's cost $\kappa(\varepsilon, \delta)$ is the number of model trainings.

1. **Spot-Checking:** The number of spot-checks is deterministic.
$$k = O(1/\varepsilon^2 \cdot \log(1/\delta))$$

2. **MSE Estimation:** $|M| = O(1/\varepsilon^2 \cdot \log(1/\delta))$.

Thus, the total cost is $O(1/\varepsilon^2 \cdot \log(1/\delta))$. Treating $\delta$ as constant yields $\kappa(\varepsilon, O(1)) \in O(1/\varepsilon^2)$.

$\square$

## E Verifying Attributions for Multiple Test Points or Outputs

Our primary protocol (Algorithm 1) and its analysis (Theorem 1) are presented for verifying attribution with respect to a single model output function $f$. However, a common and practical requirement is to obtain attributions for a model's behavior across multiple distinct scenarios, such as its predictions on $Z$ different test points, or changes in $Z$ different output logits. This gives rise to a set of $Z$ model output functions, $\{f_z\}_{z=1}^Z$, where each $f_z : \{\pm 1\}^N \to [-b, b]$. For each $f_z$, there is a corresponding empirical influence attribution $\Phi(S, z)$ and an optimal linear prediction residual $B_{\geq 2, z}$.

The goal is to extend our verification framework such that the Verifier V can, with overall confidence $1 - \delta$, simultaneously accept $Z$ attribution vectors $\{\mathbf{a}'_z\}_{z=1}^Z$ from the Prover P if and only if $\text{err}(\mathbf{a}'_z, \Phi(S, z)) \leq \varepsilon$ for all $z \in [Z]$. This section details how our protocol can be adapted to achieve this, maintaining efficiency and the two-message structure.

The core strategy involves adjusting the internal confidence parameters of the statistical estimation subroutines using a union bound. If the overall desired failure probability for the entire set of $Z$ verifications is $\delta$, then the failure probability allocated to any critical estimation step concerning an individual function $f_z$ must be reduced to $\delta' \approx \delta/Z$. This tightening of per-instance confidence directly impacts the sample complexities, but only logarithmically in $Z$.

We formalize this extension with the following theorem:

**Theorem 3** (PAC-Verification for Multiple Output Functions). *Assume that for each $z \in [Z]$, the model output function $f_z : \{\pm 1\}^N \to [-b, b]$ for some constant b. For any $\varepsilon \in (0, 1)$, and $Z \geq 1$, there exists an $(\varepsilon, \delta)$-PAC verifier for the set of $Z$ empirical influence operators $\{\Phi(S, z)\}_{z=1}^Z$. The Verifier's expected cost function $\kappa_Z$ (number of local model trainings) satisfies $\kappa_Z \in O((1/\varepsilon^2) \cdot \text{polylog}(Z) \cdot \text{polylog}(1/\delta))$. The interactive protocol requires only two messages.*

*Proof Sketch for Theorem 3.* The proof adapts the logic of Theorem 1 by incorporating the union bound across the $Z$ output functions. Let $\delta_0 = \delta/(4Z)$ be the target failure probability for individual statistical estimation steps (like residual estimation or MSE estimation for a single $f_z$) and for the spot-checking mechanism's success in detecting a certain level of fraud for any $f_z$. The result follows immediately from the logarithmic dependence of the sample complexities on $\frac{1}{\delta}$ in each step. $\square$

## F Why PAC-Verification and not Delegation of Computation?

As first mentioned in section 2, one might initially consider employing general-purpose cryptographic protocols for delegation of computation or verifiable computation to ensure the Prover P performed the expensive attribution computation correctly. However, such approaches are insufficient for the

verification goal central to our work. First of all, computational overhead of the cryptographic machinery is typically massive, and in our setting, the Prover has already pushed the limits of their feasible computations. Second, cryptographic protocols typically do not give guarantees for the *statistical quality* of the output. In particular, whether the resulting attribution scores $\mathbf{a}'$ are indeed $\varepsilon$-close to the optimal scores $\Phi(S)$.

For instance, if the Prover uses skewed, unrepresentative, or otherwise low-quality subsets for its internal estimation process (whether maliciously or inadvertently), a general-purpose delegation protocol might still verify the computation's execution correctly based on those flawed inputs, offering no protection against a poor-quality final result in terms of predictive accuracy. Note, the Verifier cannot supply the inputs due to computational constraints. A practical example of when the use of flawed inputs might occur is if the Prover itself obtains trained models from a public ledger or otherwise untrusted source.

Thus, when the Prover's computational steps are *correct*, standard cryptographic verification procedures would *not* necessarily identify bad attributions. In other words, it does not inherently assess whether the output meets a statistical benchmark defined relative to the underlying data generating process or the optimal achievable performance. The PAC-verification framework of Goldwasser et al. (2021) is designed precisely to handle verification of approximate correctness of the *outcome* relative to a statistical ideal (here, minimizing MSE), rather than merely the integrity of the computational steps taken.

