# OpenReview forum: "Efficiently Verifiable Proofs of Data Attribution"
_NeurIPS.cc/2025/Conference — NeurIPS 2025 poster_

### Official Review · Reviewer_Hayo · 2025-06-21

**Clarity:** 2
**Significance:** 2
**Originality:** 3
**Rating:** 4
**Confidence:** 2

**Summary:**

This paper introduces a novel interactive protocol that allows computationally limited verifiers to efficiently verify whether training data attribution vectors computed by a powerful prover are approximately optimal. The verification is grounded in a PAC framework, aiming to estimate the gap between the candidate attribution and the best possible attribution (in terms of MSE). The key technical contributions include a transformation of a previously known residual estimation algorithm into a two-message interactive protocol with spot-checking, reducing verifier cost from $O(1/\varepsilon^3)$ to $O(1/\varepsilon^2)$.

**Questions:**

See weaknesses.

**Ethical Concerns:**

["NO or VERY MINOR ethics concerns only"]

**Final Justification:**

The authors’ rebuttal address my concerns and some of the minor typos are also corrected, therefore, I will keep my score.

**Limitations:**

- No experiments or simulation support.

- Dependence on a strong determinism assumption.

- Possible limited practicality in modern deep learning settings.

**Paper Formatting Concerns:**

- Typos: "straight forward" to "straightforward", and the usage of "ϵ" and "ε" should be unified.

**Quality:**

3

**Strengths And Weaknesses:**

## Strenths
- Introduces elegant and efficient PAC-verification protocol for data attribution.

- Solid integration of Boolean harmonic analysis, noise stability, and residual estimation.

- Robust theoretical analysis with completeness, soundness, and efficiency guarantees.

## Weaknesses
- No experimental results. Given the strong practical motivation (e.g., verifying attributions in sensitive settings like data pricing or medical diagnosis), this absence is glaring. Even synthetic experiments could have helped ground the theoretical claims. Moreover, real-world GPU training often struggles to meet non-deterministic conditions, potentially undermining completeness/reliability.

- Lemma 1 assumes that erroneous samples are independent and uniformly distributed. If a malicious Prover selectively tampers with the subset that contributes significantly to noise-stable estimation, it can substantially shift $B̂_{≥2}$ within the corruption budget, breaking the ε/4 error bound.

- The symbol ρ is not explained when it first appears in the main text.

---

> ### Author Rebuttal · Authors · 2025-07-29
>
> Dear Reviewer Hayo,
>
> Thank you for your time and feedback. Below, we address the raised weaknesses and questions.
>
> **Experimental Results and Practicality**
>
> We agree that experimental validation is a natural next step. The primary contribution of this work is to establish the theoretical foundations for efficiently and verifiably proving data attribution, with formal guarantees of completeness and soundness against powerful, malicious Provers. Given focus on provable security and verifiability, which is difficult to simulate meaningfully, we deferred a full empirical study to future work.
>
> Regarding the non-determinism in GPU training, we discussed in the paragraph titled "Equivalence Checking Procedure" (Lines 536-543) that, while the theorems assume perfect equivalence checking procedure, the check-equiv procedure can be instantiated practically by, for instance, comparing model outputs or final loss values on a held-out set, rather than requiring a bit-for-bit identity of model weights. This accommodates the realities of modern deep learning frameworks, and any error in equivalence checking can be wrapped into the probabilities for soundness and completeness via union bounds.
>
> **On the conditions in Lemma 1**
>
> We would like to clarify that the robustness guarantee of Lemma 1 is stronger than the review suggests. The lemma does not assume that bad samples are chosen uniformly or independently by the cheating Prover.
> Instead, our analysis considers the adversarial setting, where the Prover can corrupt any $m$ function evaluations of its choosing, in order to to most impact the Verifier's estimate (see Line 571: "...an adversary corrupts up to $m$ ... of the $n$ function evaluations..."). The proof in Appendix D.1 is constructed specifically to bound the worst-case error under these adversarial, non-uniform corruptions. Therefore, the scenario described by the reviewer is precisely the threat model that Lemma 1 is proven to be robust against.
>
> **Typos**
>
> Thank you for your careful reading and help with notation and typos. We will be sure to make the edits in the final version, especially defining the symbol $\rho$ at the right time.
>
> Thank you again for your time and valuable comments.
>
> Sincerely,
>
> Authors

---

### Official Review · Reviewer_on4T · 2025-06-29

**Clarity:** 3
**Significance:** 2
**Originality:** 2
**Rating:** 4
**Confidence:** 1

**Summary:**

The paper addresses the challenge of verifying data attributions in machine learning models, particularly in scenarios where a computationally powerful entity (the Prover) provides attributions to a resource-constrained party (the Verifier). The authors propose an interactive verification protocol to ensure that the attributions are "good" (i.e., close to optimal) without requiring the Verifier to perform the computationally expensive task of computing the attributions themselves.

**Questions:**

Add a flowchart or simplified pseudocode in the main text to visually summarize the interactive protocol (beyond Algorithm 1 in the appendix). This would help readers quickly grasp the key steps (e.g., challenge setup, spot-checking, residual estimation).

**Ethical Concerns:**

["NO or VERY MINOR ethics concerns only"]

**Limitations:**

The paper acknowledges that retraining-based attribution methods (like datamodels) are computationally expensive, creating a trust gap between powerful entities (e.g., corporations) and resource-limited users (e.g., individuals or small labs). This is a key motivation for the work.

**Quality:**

2

**Strengths And Weaknesses:**

Strengths:
- Well-Structured: The paper is logically organized, with clear motivation, problem formalization, protocol description, and proofs.
- Good Intuition: The authors provide intuitive explanations (e.g., the "naive verification fails" example) before diving into technical details.
- Full Proofs: The main text outlines key ideas, while detailed proofs are deferred to appendices, improving readability.

Weaknesses:
- The heavy reliance on Fourier analysis and Boolean function notation might make the paper less accessible to readers without a strong theoretical background.
- While Algorithm 1 is detailed, a high-level pseudocode summary in the main text could improve clarity for practitioners.

---

> ### Author Rebuttal · Authors · 2025-07-29
>
> Dear Reviewer on4T,
>
> Thank you for your time and feedback!
>
> In regards to the following weakness statement in your review:
>
> > While Algorithm 1 is detailed, a high-level pseudocode summary in the main text could improve clarity for practitioners.
>
> We would like to clarify that Algorithm 1 does have a pseudocode/summary in the main text, at the beginning of section 4. If we have a misunderstanding, please let us know and we are happy to clarify further.
>
> Thanks again for your review.
>
> Sincerely,
>
> Authors

---

### Official Review · Reviewer_BYJZ · 2025-07-01

**Clarity:** 3
**Significance:** 3
**Originality:** 3
**Rating:** 5
**Confidence:** 3

**Summary:**

Suppose that we train an ML model on an $N$ size dataset, and we would like to know how much each data element (or subset of elements) influenced the training. This is modeled by a function $f \colon \\{-1,1\\}^N \rightarrow \mathbb{R}$, where each $x \in  \\{-1,1\\}^N$ encodes a subset ($x_i = 1$ iff the $j$’th element is included in the subset), and this function can be estimated for every $x$ by retraining the model on the subset $x$ and then computing the prediction quality or test error of the new model. The problem is that computing this function is very expensive (each subset requires a fresh training), and we would like to compute an approximation of $f$, hopefully using only a few retrainings. The work of Saunshi et al. (2022) showed how to compute a linear model $a$ such that $f(x) \approx <a,x>$, where the number of retrainings to obtain $a$ is only $\approx 1/\epsilon^3$, which is independent of $N$ ($\epsilon$ is the required accuracy measured by the MSE distance from the optimal linear model). This is done using Residual Estimation.


This paper considered a model where we have two entities: a prover, which has more computational resources, and the verifier, which is more restrictive but still would like to get the estimation of $f$. The problem arises when the prover is not necessarily honest, so the verifier must have a way to check whether the linear model $a$ produced by the prover is indeed close to optimal. The main result of this paper is to provide a 2-message interactive protocol between the prover and the verifier, where the verifier, using only $\approx 1/\epsilon^2$ retrainings (and not $\approx 1/\epsilon^3$ as in the non-interactive version of Saunshi et al. (2022)), can catch an unhonest prover with high probability. The solution relies on the same Residual Estimation of Saunshi et al. (2022): The verifier asks the prover to perform the $1/\epsilon^3$ retrainings, but checks only $\epsilon$ fraction of them by doing the retrainings itself (where the exact $\epsilon$ fraction is unknown to the prover). If the prover lied in more than $1/\epsilon$ out of the $1/\epsilon^3$ requests, the verifier will catch it with high probability. Otherwise, there could be at most $1/\epsilon$ wrong values, so the main technical contribution of this paper is to prove that the approach of Saunshi et al. (2022) is error-robust against such an amount of errors.

**Questions:**

No questions

**Ethical Concerns:**

["NO or VERY MINOR ethics concerns only"]

**Final Justification:**

I had no questions for the authors, and the other reviews were also supportive, so I'm keeping my positive score.

**Limitations:**

yes

**Paper Formatting Concerns:**

No formatting concerns

**Quality:**

3

**Strengths And Weaknesses:**

I haven’t read the robustness proof that appears in the appendices (Lemma 1, page 15), but assuming it is correct, I think it is a nice paper that provides a nice solution to a well-motivated problem while addressing a real technical challenge (the robustness of the residual estimation). Therefore, I recommend acceptance.

---

> ### Author Rebuttal · Authors · 2025-07-29
>
> Dear Reviewer BYJZ,
>
> Thank you for your time and effort in completing your review. We appreciate the vote for acceptance.
>
> Sincerely,
>
> Authors

---

> > ### Comment · Reviewer_BYJZ · 2025-08-05
> >
> > Thank you for your response.

---

### Official Review · Reviewer_cTDk · 2025-07-02

**Clarity:** 4
**Significance:** 3
**Originality:** 4
**Rating:** 5
**Confidence:** 4

**Summary:**

This paper studies the "verifiable" problem for data attribution, an important growing field in data-centric AI. Specifically, the paper defines the verifying problem in data attribution and proposes an "efficient" interactive protocol that is PAC-verifiable.

**Questions:**

1. Can the authors discuss what is the role of the final attribution score $a'$ produced by the prover? Since, at the end of the day, if the prover is honest/passes the spot-checking, the verifier will use the provided models and curate the final attribution scores using the residual estimation algorithm. It seems to me that the attribution score $a'$ provided by the prover is useless in this sense.
2. (**Weakness 1**) I suggest that the authors can elaborate and clarify the notion of efficiency used in this paper more explicitly.
3. (**Weakness 2**) I understand that due to the space limits, no related works are presented in the paper. I suggest that (if published) the authors can add a dedicated related work section to carefully discuss the potential impacts and some design choices to help justify and position the paper in the literature.

**Ethical Concerns:**

["NO or VERY MINOR ethics concerns only"]

**Final Justification:**

I maintain my positive score, and no further concerns are raised after reading all the discussions with other reviewers.

**Limitations:**

While no specific limitations were discussed in the paper, the reviewer does not have additional concerns on potential limitations.

**Quality:**

3

**Strengths And Weaknesses:**

**Strengths**
1. With the growing interest in data attribution, with its potential societal impact on data compensation (or data economy), the problem studied in this work is indeed important and well-motivated.
2. The writing is professional, consistent, and self-contained.
3. The proposed algorithm and interactive protocol are clear and well-motivated (with the appropriate introduction to the prior works), which makes them easy to follow overall.

**Weaknesses**
1. If my understanding is correct, the notion of "efficiency" is the number of retrainings required. It is, hence, a bit misleading to say that the "cost" is independent of dataset size $N$. I suggest that the authors can elaborate and clarify this point in the next iteration.
2. The scope of the present work is a bit restricted and not justified: for instance, the current vast data attribution literature has moved beyond retraining-based datamodel style methods to explore settings such as "single-model TDA" via, e.g., dynamic methods. I think the design of the present work is largely due to the prior work by Saunshi et al., which makes sense, but it'll be helpful to discuss this more in detail.

**Writing**
Nothing really significant. Just want to point out:
1. Page 13 can probably be handled more elegantly..

---

> ### Author Rebuttal · Authors · 2025-07-29
>
> Dear Reviewer cTDk,
>
> Thank you for your kind and thoughtful review!
>
> **Notion of "Efficiency"**
>
> Indeed, you are correct. We definitely didn’t mean to be misleading and will clarify/correct it in the final version. Currently, our use of "efficiency" refers to the Verifier's sample complexity (the number of retrainings), which is itself independent of the dataset size $N$. The cost of each individual retraining does depend on $N$, and we will make this distinction explicit in the final version.
>
> **Scope/Relation to Other Methods**
>
> We feel that focusing on datamodels and empirical influences in the context of predictive attribution will always be a worthwhile focus because they are the provably optimal linear attributions for minimizing predictive Mean Squared Error (MSE) (which is our metric). This is proved by [1].  Verifying w.r.t. empirical influence / datamodels thus addresses the fundamental question of whether a party has computed the best possible linear explanation. We agree that while related work is discussed as necessary throughout the body of the paper, a dedicated section would also be valuable, especially mentioning other lines of work like single model TDA. We will add this for the final version.
>
> **Prover's Attribution Score**
>
> The Prover's attribution vector $a^{\*}$ is very important. The core of the Verifier's protocol is to check if the error of this submitted particular $a^{\*}$ is close to the optimal error (Algorithm 1). Without $a^{\*}$ from the Prover, there would be no candidate solution for the Verifier to check.
>
> Thank you again for your valuable comments.
>
> Sincerely,
>
> Authors
>
>
> -----
>
> **References**
>
> [1] Saunshi, N., Gupta, A., Braverman, M., and Arora, S. (2022). Understanding influence functions and datamodels via harmonic analysis. arXiv preprint arXiv:2210.01072.457

---

### Official Review · Reviewer_fc4o · 2025-07-03

**Clarity:** 4
**Significance:** 3
**Originality:** 3
**Rating:** 6
**Confidence:** 4

**Summary:**

This paper proposes a technical solution to the trust problem centered around data attribution vectors computed from datamodels. In this context, datamodels refer to mappings that quantify the relationship between subsets of a training dataset and the resulting performance of a model trained on those subsets. For example, given a fixed training set of size $N$, we can define a mapping $f: \\{\pm 1\\}^N \to [0,1]$, where the selection vector $x \in \\{\pm 1\\}^N$ indicates which data points are included in the subset and $f(x)$ gives the final 0/1 loss of the classifier trained on that subset.

Current approaches compute data attribution vectors from datamodels. They are defined as the linear coefficients $a \in \mathbb{R}^N$ of the optimal linear approximation to $f$ with respect to an inner product. For simplicity, we work in the inner product space induced by the uniform distribution over $\\{\pm 1\\}^N$. The analysis extends essentially unchanged to the case of i.i.d. products of $p$-Bernoulli distributions. These vectors have important practical applications, such as data pricing, allowing one to compensate individual data points within a training set according to their “influence” on the final performance of the model.

Evaluating the datamodel $f$ on each training subset $x$ is extremely expensive, and therefore computing honest data attribution vectors creates significant asymmetry and trust issues. Only a few parties have the resources to perform these computations, while most others must rely on, and therefore trust, this small group of powerful actors. The paper addresses this asymmetry by introducing an interactive protocol in which the Verifier can check the *optimality* of the Prover’s data attribution vector with much less computational cost. Here, the Verifier’s computational cost, defined to be the number of datamodel evaluations, is shown to be *independent* of the dataset size $N$.

A key technical insight comes from previous work by Saunshi et al., who observed that computing certain Fourier coefficients of the residual $g(x) = f(x) - \langle a, x\rangle$, given $a$ is much cheaper than computing the attribution vector from scratch. The novelty of this paper lies in formulating the trust problem as a PAC verification task and offloading the computation of the residual’s Fourier coefficients to the Prover. With the spectral computation pushed to the Prover, the Verifier only needs to perform random spot checks on the Prover’s results to verify their validity. The offloading reduces the Verifier’s computational cost from $1/\epsilon^3$ to $1/\epsilon^2$ (ignoring other parameters of the interaction), where $\epsilon$ denotes the (additive) suboptimality of the Prover’s candidate attribution vector relative to the optimal one.

**Questions:**

1. Could the authors discuss the shortcomings of the current definition of Verifier’s computational cost?
2. What general assumptions on the equivalence checking procedure are sufficient to preserve the PAC verification guarantees?
3. How sensitive are the attribution vectors to the choice of $p$ in the $p$-Bernoulli marginal distribution over selection vectors? For example, are there empirical studies examining how the ranking of data point influence varies with $p$?

**Ethical Concerns:**

["NO or VERY MINOR ethics concerns only"]

**Final Justification:**

Rating remains the same. Reasons are provided in my response to the authors' rebuttal.

**Limitations:**

yes

**Quality:**

4

**Strengths And Weaknesses:**

### Strengths

1. **Compelling narrative and motivation.** The paper presents a creative and well-motivated application of interactive proofs, clearly explaining how asymmetry in computational cost creates trust problems in data attribution, and highlighting the need to resolve them for practical applications such as data pricing. Formulating this trust problem as an interactive protocol feels natural and provides an elegant framework for a real, practical issue.
2. **Insightful use of tools from theoretical computer science.** The paper elegantly combines Fourier analytic ideas from Saunshi et al. and techniques from interactive proofs to arrive at a quantitative improvement. Because random spot checks are sufficient to detect the Prover’s deviations with high probability, the Verifier can shift the computational burden of residual estimation to the Prover, an approach very much in the spirit of probabilistically checkable proofs (PCPs).
3. **Clear, concise, and precise exposition.**  The paper is exceptionally well-written, presenting the technical content in an accessible way without compromising rigor. Moreover, the narrative and technical tools are tightly integrated, resulting in a seamless presentation that effectively connects the motivation to the methodology.

### Weakness

1. **Missing assumptions on equivalence checking.** The paper lacks precise details about the assumptions required for the equivalence checking procedure. I assume all PAC verification guarantees hold if model training is deterministic, with CheckEquiv(a_1, a_2) returning true if and only if a_1 = a_2. However, if there is significant randomness or noise in training outcomes, completeness and soundness guarantees may fail to hold. For example, if CheckEquiv always passes any input pair, the protocol would fail to satisfy soundness guarantees. A clearer specification of the conditions under which equivalence checking ensures both soundness and completeness would strengthen the paper.

2. **Other choices of computational cost for the Verifier.** The claimed independence of Verifier’s cost from $N$ is somewhat misleading. While the number of datamodel evaluations required for the protocol is indeed independent of $N$, the cost of each individual evaluation can grow with $N$. For example, even with a fixed inclusion probability $p$, the expected training subset size scales with $pN$, and the cost of training a model on these subsets may increase accordingly. Moreover, in high-dimensional settings where the data point dimension $d$ scales linearly or superlinearly with the number of samples $N$, even parsing a single data sample can scale with $N$. In this sense, a more budget-relevant accounting of computational costs may not be truly independent of $N$. Clarifying these caveats and considering alternative, reasonable notions of the Verifier’s computational cost would strengthen the paper.

---

> ### Author Rebuttal · Authors · 2025-07-29
>
> Dear Reviewer fc4o,
>
> Thank you for your kind and thoughtful review!
>
> **About the Notion of "Efficiency"**
>
> Your point is correct. We certainly don't mean to mislead and will clarify it in the final version. Currently, our use of "efficiency" refers to the Verifier's sample complexity (the number of retrainings), which is itself independent of the dataset size N. The cost of each individual retraining does of course depend on N, and we will make this distinction explicit in the final version of our paper.
>
> **Non-determinism in GPU training**
>
> We would like to point out that we did discuss this, in the paragraph titled "Equivalence Checking Procedure" (Lines 536-543) of the submission. While the theorems as stated do assume a perfect equivalence checking procedure, the check-equiv procedure can be instantiated practically by, for instance, comparing model outputs or final loss values on a held-out set, rather than requiring a bit-for-bit identity of model weights. Any error in equivalence checking can be wrapped into the probabilities for soundness and completeness via union bounds.
>
> Regarding your question:
>
> > How sensitive are the attribution vectors to the choice of $p$ in the $p$-Bernoulli marginal distribution over selection vectors? For example, are there empirical studies examining how the ranking of data point influence varies with $p$?
>
> In general, the choice of $p$ does affect the attributions. Different values of $p$ were used and studied in, for example, [1].
> However, due to high linearity observed in practice on typical benchmark settings (e.g., ResNet-18 trained on CIFAR-10), the values of attributions are not "very" sensitive to $p$.
>
> Thanks again for your kind review!
>
> Sincerely,
>
> Authors
>
> -----
>
> **References**
>
> [1] Ilyas, A., Park, S. M., Engstrom, L., Leclerc, G., and Madry, A. (2022). Datamodels: Predicting
> predictions from training data. arXiv preprint arXiv:2202.00622.

---

> > ### Comment · Reviewer_fc4o · 2025-08-03
> >
> > Thank you for the clarification and for being receptive to my comment about notions of efficiency for the interactive protocol. Since my questions were only minor clarifications, they did not affect my (strongly positive) assessment of the paper. Therefore, my rating remains unchanged. Great work!

---

### Decision · Program_Chairs · 2025-09-17

**Decision:**

Accept (poster)

**Comment:**

Estimating data subset influence on an ML model is computationally expensive due to repeated retraining. Building on Saunshi et al. (2022), who proposed an efficient linear approximation, this work introduces an interactive protocol where a weak verifier outsources this task to a powerful but untrusted prover. The verifier spot-checks a small fraction of the prover's computations. The paper's main technical contribution is proving that the approximation method is robust against a small number of errors, which allows the verifier to efficiently detect a dishonest prover. All reviewers are supportive, and the paper would make a great addition to NeurIPS.